

# Observation of nocturnal NO$_3$ during vehicular activities in the medium sized city of Calicut in coastal India

Kuttoth Suhail[1], Ramachandran Arun[1*], Shreya Joshi[3], John Shebin[1], Saseendran Aiswarya[1], Pakkattil Anoop[1], Viswanath Deepa[2], Ravi Varma[1]

[1]Department of Physics, National Institute of Technology Calicut, Calicut, 673601, India
[2]Depatment of Physics, VTM NSS College, Trivandrum, 695503, India
[3]Optind Solutions Pvt. LTD., Unit 11, Technology Business Incubator, National Institute of Technology Calicut, Calicut, 673601, India

*Correspondence to*: R. Arun (arunrpkd@gmail.com)

**Abstract.** Nitrate radical (NO$_3$) is the most important nocturnal oxidant in urban atmosphere. The city of Calicut (Kozhikode) is a medium sized urban location in India. One of the bus terminals at Palayam part of the city [11.2495° N, 75.7842° E] is adjacent to the vegetable sorting facilities cum market, both of which have intense activities by buses and trucks from about 3 AM till about 8 PM on all working days. We report preliminary measurements observing NO$_3$ on five nights during a weeklong measurement campaign in the autumn of 2018. Measurements were made between 10 PM and 6 AM, and focus was when diesel vehicles were found idling along about half a km stretch during 3 – 6 AM while the loading/unloading of vegetables at sorting facility happens. Incoherent Broadband Cavity Enhanced Absorption Spectroscopic technique in open-path configuration (OP-IBBCEAS) was employed for measurements. The instrument was installed 8.5 m above the ground level over the east wing of Palayam bus terminal building adjacent to the vegetable market. The 285 cm long optical resonator was arranged on a custom-made aluminium profile platform 1 m high. The stability of the instrument for the entire period of measurement was excellent, and high NO$_3$ mixing ratios with levels exceeding several hundred pptv were observed during early morning hours when heavy vehicles were idling. The highest NO$_3$ mixing ratio observed was (497 ± 140) pptv during one of the nights. The fit uncertainty, which was considered as the uncertainty in retrieved concentration, was found to increase with increased aerosol loading. The uncertainty for a spectral averaging interval of 10 min was recorded as ~20 pptv and ~100 pptv during the lowest and the highest aerosol loading events respectively.

## 1 Introduction

The photochemistry of the atmosphere induced by sunlight triggers the formation of various free radicals, like nitrate and hydroxyl radicals and other vital trace gas species [Bohn et al., 2005; Monks et al., 2005]. The day and night time chemistry of the atmosphere are different and the in situ monitoring of the radical species are essential because of their high reactivity and short lifetimes [Khan et al., 2015; Larin et al., 2014; Geyer et al., 2003]. The adverse health impacts of these trace gases



have also been recognized as far reaching [Sarnat, 2016]. Being a dominant night-time oxidant, $NO_3$ triggers the formation of other trace species like nitric acid and peroxy radicals [Wayne et al., 1991; Atkinson, 1991]. It reacts with various hydrocarbons and enhances VOC oxidations [Khan et al., 2015]. The presence of $NO_3$ is an indication of VOCs (source) and $N_2O_5$ (sink) in the atmosphere.

Non-invasive techniques for trace gas monitoring with high sensitivity and time resolution are available of which high finesse optical cavity methods like Cavity Enhanced Absorption Spectroscopy (CEAS) and Cavity Ring Down Spectroscopy (CRDs) are well established [Zheng et al., 2018; Nowakowaski et al., 2009]. These spectroscopic instruments are suitable for both laboratory and field measurements of the trace species as well as particulate matter with high accuracy [Wojtas et al., 2014; Ball et al., 2004; Ling et al., 2015]. Many trace gas measurement studies have been carried out around the world using incoherent broadband CEAS technique (IBBCEAS), since the time it has been first proposed [Fiedler et al., 2003]. A key advantage of IBBCEAS is multiple gas detection capability with high spatial and temporal resolution [Ruth et al., 2014], an important feature that makes the IBBCEAS popular and relevant methodology for *in-situ* trace gas monitoring. IBBCEAS has been used in wide spectral bands ranging from ultraviolet (UV) to near infrared (NIR) for various applications. To name a few: monitoring of molecular iodine [Dixneuf et al., 2008; Johansson et al., 2014], glyoxal [Washenfelder et al., 2008], IO and OIO [Vaughan et al., 2008], and $NO_2$ [Chandran et al., 2017; Langridge et al., 2006; Triki et al., 2008; Liang et.al., 2017] using the visible spectral range; HONO [Nakashima et al., 2017; Wu et al., 2012; Gherman et al., 2008; Duan J. et.al., 2015], BrO [Chen et al., 2011], and gaseous elemental Hg [Darby et al., 2012] using the UV spectral range; different VOCs [Orphal, and Ruth, 2008; Denzer et al., 2009], and natural gas components [Prakash et al., 2018] using the near IR spectral range have been reported. Open-path IBBCEAS (OP-IBBCEAS) measurements in simulation chambers were reported for $NO_2$ and $NO_3$ [Venables et al., 2006; Varma et al., 2009] and recent field measurements using the same configuration has reported a detection sensitivity of ~40 pptv for $NO_3$ even in the presence of aerosol loading [Suhail et al., 2018].

Ambient measurements of $NO_3$ have been widely conducted in USA, China and Europe. [Langridge et al., 2008; Kennedy et al., 2011; Wang et al., 2017, Wang et al., 2013]. However, few such studies have been reported from urban locations of India. In this study, we describe the use of OP-IBBCEAS for *in-situ* measurements of nocturnal $NO_3$ radical in a medium sized coastal city of Calicut (Kozhikode) in southern state of Kerala, India. The location of the measurements was chosen where heavy vehicles such as buses and trucks were operated heavily, and in idle conditions at times. The instrument was installed 8.5 meter above ground level on the east wing of the oldest of the three bus terminals [11.2495° N, 75.7842° E] of the Palayam part of the city that is also adjacent to the main vegetable sorting cum market facility, both of which are busy from approximately 3 AM till about 8 PM. Measurement campaign was chosen for a one-week period during 31$^{st}$ October – 10$^{th}$ November, 2018, and the presence of $NO_3$ at high mixing ratios were observed for five nights. The instrument stability was very good throughout the campaign with minimal optical realignment and optimizations. The experimental setup, calibration, measurements and results are discussed in the following sections.



## 2 Experimental

The OP-IBBCEAS instrument was deployed over the Palayam bus terminal on top of its eastern wing of the main terminal. The location is proximate to air pollution from various human and vehicular activities. Adjacent to the bus terminal is a vegetable sorting cum market where supplies are initiated to other parts of the district of Calicut. Hence, high traffic of

trucks, idling during loading and unloading of vegetables, add to the pollution in addition to the buses moving in and out of the terminal. The Calicut city railway station is 0.5 km away from this site and is also very close to a key business street of the city. Another air quality monitoring station (observatory), also operated in this same location, was making hourly averaged measurements of $NO_x$, $O_3$, $PM_{10}$, $PM_{2.5}$, RH and temperature. Since, $NO_3$ formation is triggered by the presence of $NO_2$ and $O_3$, and OP-IBBCEAS measurement sensitivity depends on aerosol loading [Ruth et al., 2014], these concurrent

measurements were valuable additions to this measurement campaign. Figure 1(a) shows the Google map image of the location and Fig. 1(b) that of the bus terminal.

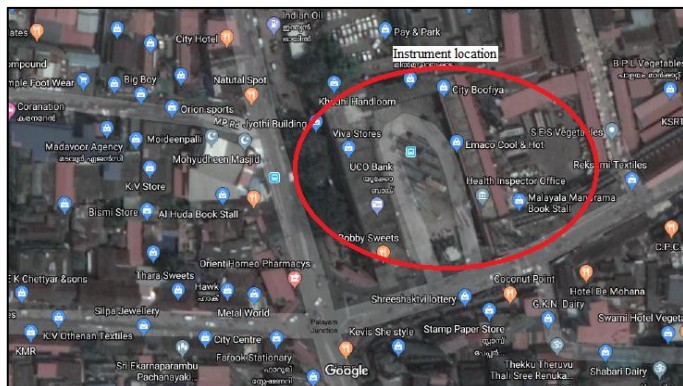
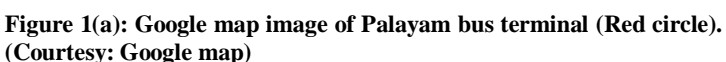
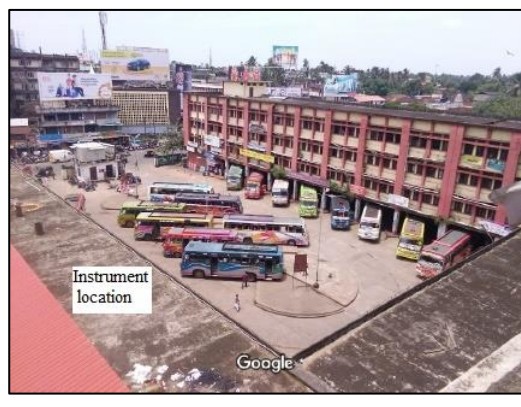

**Figure 1(a): Google map image of Palayam bus terminal (Red circle).**
**(Courtesy: Google map)**
**Figure 1(b): View of the main Palayam bus terminal from the instrument location (Pic courtesy: Google map)**

The OP-IBBCEAS instrument was placed on the rooftop of the east wing of the bus terminal on a single aluminium profile platform 4 m long and 1 m high. The schematic depiction of the experimental setup is shown in Fig. 2. The optical resonator was achieved by using a pair of high reflective (HR) dielectric mirrors with a radius of curvature of 3 m and diameter of 25.4

20   mm (Layertec GmbH). The company specified reflectivity of the mirror is > 0.999 in the 620 -720 nm spectral range. The HR mirror pair was mounted inside a custom made mirror mounts that have provision for purge gas inlet as well as insertion of a low-loss optic (used for the mirror reflectivity calibration). A Xe-arc lamp that works with an optical power of 150 W was used as the light source. L1 and L2 are lenses used to collimate the diverging beam from the arc lamp and to focus the beam at the centre of the cavity. Plane aluminium mirrors (M1 and M2) of 50 mm diameter were used to direct the beam

into the optical resonator. A 650 nm long-pass (Andover 650FS80-50) and a 700 nm short-pass (Thorlabs) filter combination were used to spectrally filter the light from the arc lamp to match mirror reflectivity range. Along with this, a



colour glass filter (Newport FSQ-KG5) was used to eliminate the IR emissions from the lamp. A TE-cooled spectrometer with CCD array (Ocean Optics, model QE Pro) featuring high signal-to-noise ratio was used as the detector, with a focusing optic (L3) collecting the cavity signal. A He-Ne laser (JDS Uniphase, USA) was used for preliminary cavity alignment, with two aluminium mirrors (M3 and M4) guiding the laser beam through the cavity.

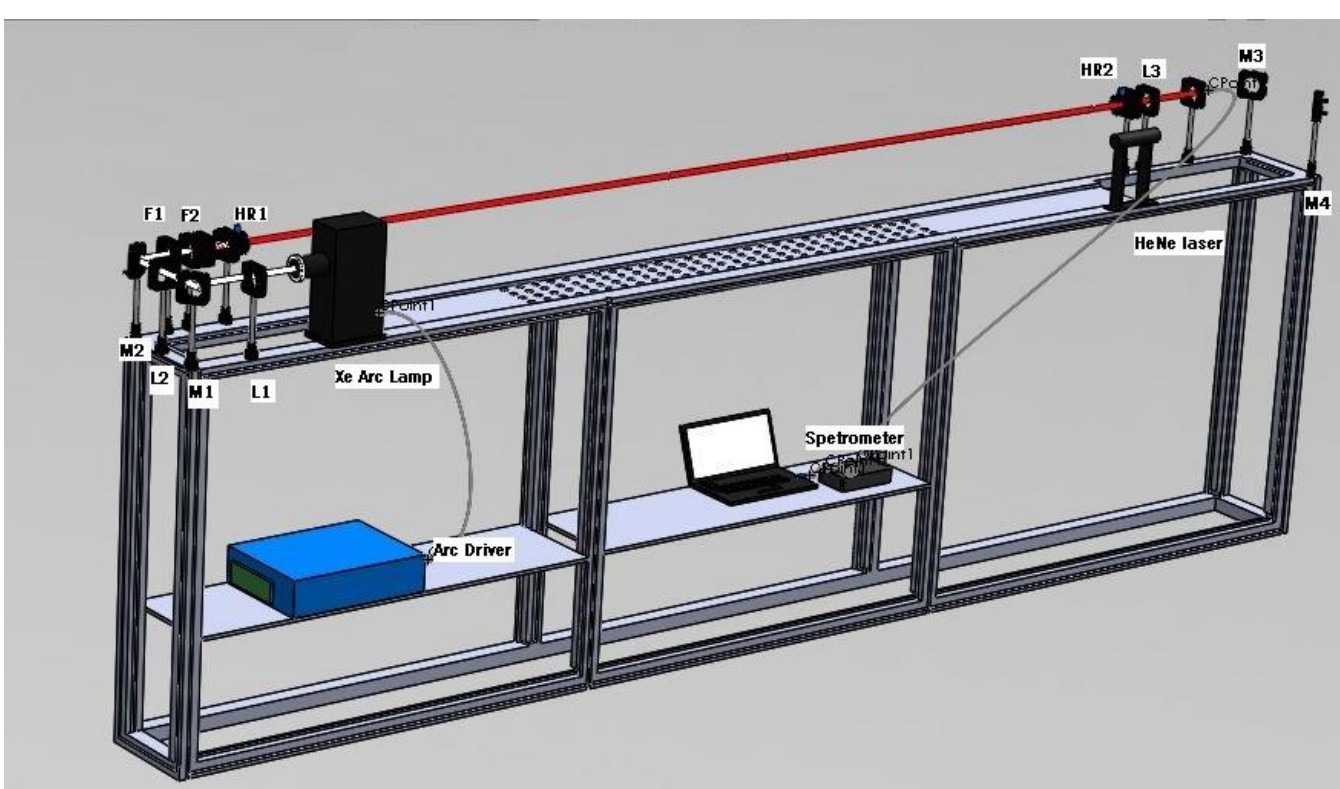

**Figure 2: Schematic drawing of the OP-IBBCEAS setup on aluminium profile platform. HR1 and HR2 are high-reflectivity cavity mirrors. M1, M2, M3 and M4 are beam steering aluminium mirrors. L1, L2 are collimating lenses and L3 is the focusing lens to the fibre. F1 is the 650 long pass and 700 short pass filter combination and F2 is the the IR filter.**

The length between two cavity mirrors was 285 cm. The cavity mirrors were purged by nitrogen gas at a rate of 100 mL/min for cavity mirror protection. The resolution of the spectrometer was determined to be 1.8 nm based on the FWHM of He-Ne laser line at 632.8 nm, and a Mercury-Argon calibration source (Ocean Optics, HG1) ascertained its wavelength calibration. A polyvinyl chloride (PVC) pipe 5 cm in diameter was connected between the cavity mirrors and $N_2$ was filled for

15  acquisition of the reference spectrum ($I_0$) prior to measurements each night. A steady flow of nitrogen was maintained in atmospheric pressure during these measurements, which was achieved with the help of a manometer U-tube. The instrument calibration is the determination of the effective mirror reflectivity of the resonator. The calibration procedure was following previous studies [Varma et al., 2009; Suhail et al., 2018] by using a pre-calibrated low-loss optical window (Layertec





GmbH) with high transmission in the spectral range of the resonator. Soon after the reference spectrum was acquired, the low-loss optic was introduced into the optical path with nitrogen as background and effective mirror reflectivity was determined.

5    After the acquisition of $I_0$ and reflectivity calibration each night, the pipe was removed so that the cavity was open to ambient air. Each spectrum collected thereafter ($I$) was recorded at 1 min interval. The IBBCEAS equation, described in Eq. (1) below was used to calculate the extinction coefficient spectra [Fiedler et al., 2003].

$$\alpha(\lambda) = \frac{1}{d}\left(\frac{I_0(\lambda)}{I(\lambda)} - 1\right)(1 - R(\lambda)) \tag{1}$$

where $d$ is the length of the resonator, $R$ is the effective mirror reflectivity, $I_0$ and $I$ are the reference cavity transmission intensity and the sample cavity transmission intensity respectively as recorded by the spectrometer. The $NO_3$ concentration was retrieved from the cavity signal by fitting each spectrum with the sum of a quadratic polynomial baseline and the product of the number concentrations and absorption cross-sections of $NO_3$ [Yokelson et al., 1994], $NO_2$ [Vandaele et al.,
1998] and water vapour [Rothman et al., 2005]. The fitting equation describes in Eq. (2) below was used [Varma et al., 2009],

$$\alpha(\lambda) = a_0 + a_1\lambda + a_2\lambda^2 + a_3\sigma_{NO3}(\lambda) + a_4\sigma_{NO2}(\lambda) + a_5\sigma_{H2O}(\lambda) \tag{2}$$

where $\alpha(\lambda)$ is the extinction coefficient of the mixed sample, calculated from Eq. (1). $\sigma_{NO3}$, $\sigma_{NO2}$ and $\sigma_{H2O}$ (from HITRAN database) are the absorption cross-sections convolved to the resolution of the spectrometer. $a_3$, $a_4$ and $a_5$ are the number densities of each species in the mixed sample and the first three terms of the equation accounts for the baseline offset of the extinction spectrum and the presence of $O_3$ in the sample.

Previous studies involving *in-situ* monitoring have reported issues with non-linear absorption of water vapour while retrieving $NO_3$ concentrations at lower resolutions when high-resolution HITRAN lines were convolved for least-square fitting [Bitter et al., 2005; Varma et al., 2009; Suhail et al., 2018]. In order to compensate for the non-linear Beer-Lambert behaviour, concentration-corrected absorption cross-section for water vapour was calculated from the average relative humidity (RH) and temperature (T) measurements obtained from the observatory. The water vapour concentration for the
same was determined from RH and T following McRae (1980). A linear least square fit algorithm developed in MATLAB environment was used for the spectral fitting. The fit uncertainty returned by the algorithm was used as a measure of detection sensitivity in these measurements.


## 3 Results and Discussions

Urban Indian atmosphere is characterized by high rate of pollution with NO$x$ from vehicular exhaust, VOCs and O$_3$, in addition to particulate matter (PM). However, little effort had been visible in the literature studying the formation and destruction of NO$_3$, the most powerful nocturnal oxidant, in Indian urban environment. Invent of sensitive *in situ* detection techniques has made observation of the same graspable. Although a comprehensive and simultaneous monitoring of several reactive species is necessary to assess dynamics and transport of NO$_3$, observation of its presence at significant levels is highly relevant. In this study, the measurements were focussed during the night-time activities of heavy/medium diesel vehicles. During the week of observation about 120 such vehicles were active during 3 – 5 AM on each market day solely for sorting vegetables, and may have been the source of pollution for our observations. The activities close to the sorting facility (next to the instrument location) is roughly limited in a stretch of half km and the vehicles are left on idle during loading/unloading of vegetables. During normal working hours, this street south of the bus terminal, and the cross road to its west have high traffic of slow moving passenger vehicles (mostly cars and two-wheeler gasoline vehicles). The NO and NO$_2$ levels between 3 – 5 AM measured at the observatory located in the same terminal showed a minimum in daily values of ~20 and ~30 µg m$^{-3}$ respectively (private communication). During daytime they peaked to ~120 and ~80 µg m$^{-3}$ respectively on an average during the campaign period. These hourly observations were not analysed further for correlations due to the lack of comprehensive VOC measurements. During the measurement period significant wind was not prevailing, hence transport of pollutants from elsewhere was held negligible (~ 0.5 ms$^{-1}$ measured by anemometer at the observatory, mostly from the West that is from the direction of the coast).

The average effective mirror reflectivity ($R$) spectrum with standard deviation from all 5 days of measurement is shown in Fig. 3. The manufacturer specified reflectivity was > 0.999 in the spectral range of 620 – 720 nm. The actual reflectivity at 662 nm was determined to be ~0.9997.

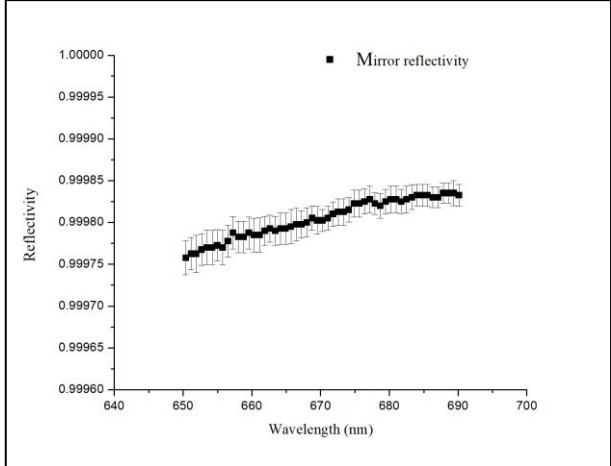





**Figure 3: Mirror reflectivity (averaged over all 5 days of measurement). Error bar represents the standard deviation at corresponding wavelengths with a relative uncertainty of $3.2 \times 10^{-5}$ at 662 nm.**

Typical transmission spectra $I_0$ (red trace) and $I$ (blue) trace are shown in Fig 4(a). The differences in the spectra are attributed to the presence of aerosol, water vapour and other atmospheric absorbing and scattering constituents. In fact, the acquisition of $I$ spectrum in the figure was made at 10:15 PM on November $5^{th}$ when the presence of $NO_3$ was detected. A mixing ratio of 40 pptv was retrieved with a fit uncertainty of 20 pptv.

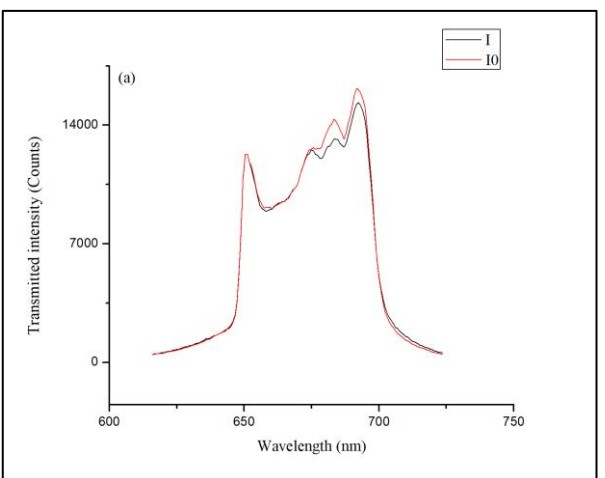
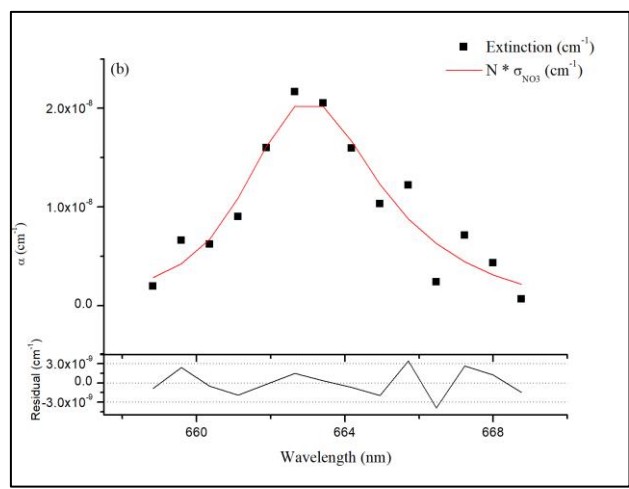

**Figure 4: (a) Typical cavity transmission signal on day 4 when aerosol loading was relatively low. The red trace is the $I_0$ spectrum and the blue trace is the $I$ spectrum when $NO_3$ was present at low levels (b) The absorption coefficient in 658 – 668 nm range showing $NO_3$ absorption alone (black square) with all other contributions (water vapour, $NO_2$ and a polynomial baseline) removed and its fit in red. Also shown is the fit residual. (The mixing ratio of $NO_3$ retrieved from this spectrum was $40 \pm 20$ pptv)**

The RH values were fluctuating around 80% levels for most of the measurement period. The water cross-section spectrum in the spectral range of analysis was corrected for a corresponding concentration to account for nonlinear Beer-Lambert's behaviour throughout the campaign. During the whole week of campaign it was noticed that the vehicular activities start around 3 AM and remain busy till about 8 PM on every working day. Data acquisition after calibration was commenced from about 10 PM each night and continued till sunrise, which is about 6 AM in the morning at this location during the measurement period. A relatively calm period was observed during the night until after 2 AM. The $NO_3$ mixing ratio was observed to jump up shortly after the activities commence in the surroundings, and trucks running in idle engines may be a major source of pollution.





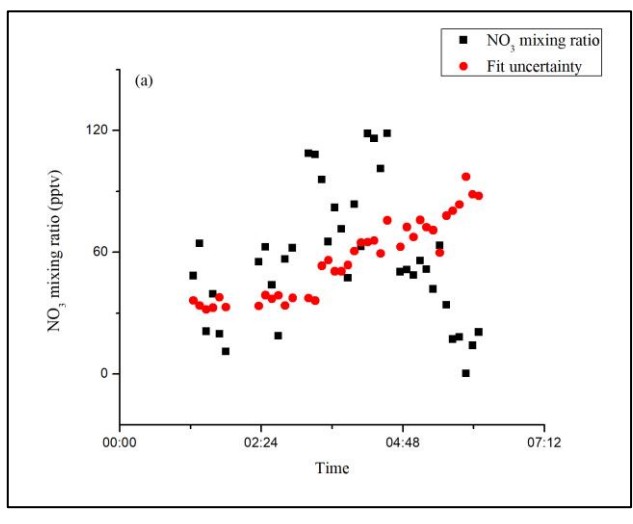
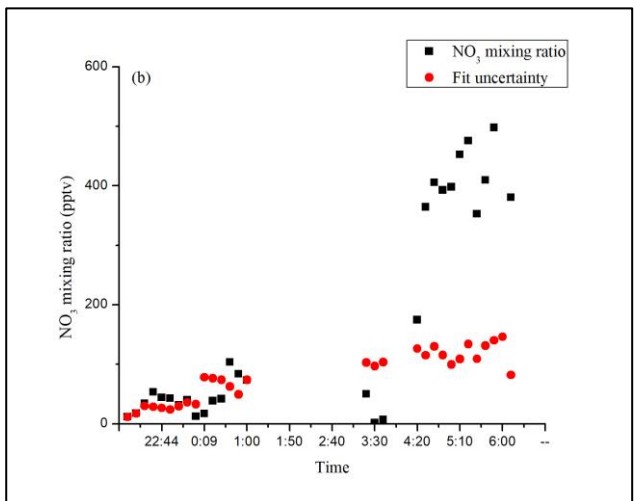

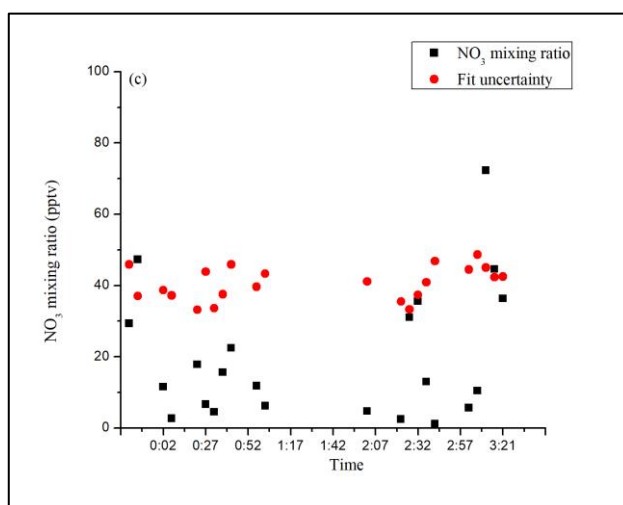
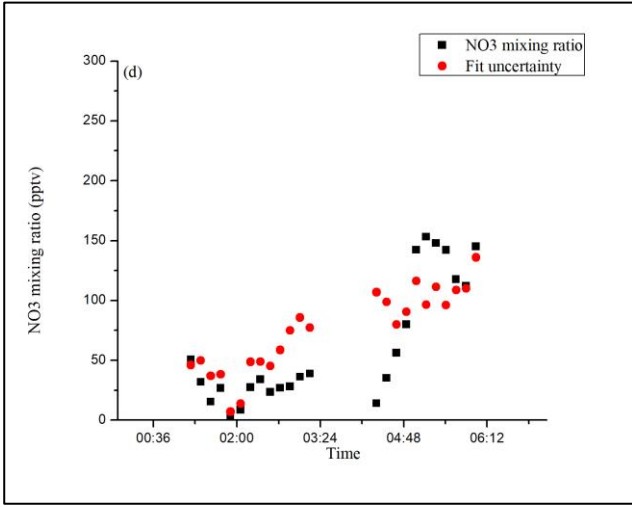

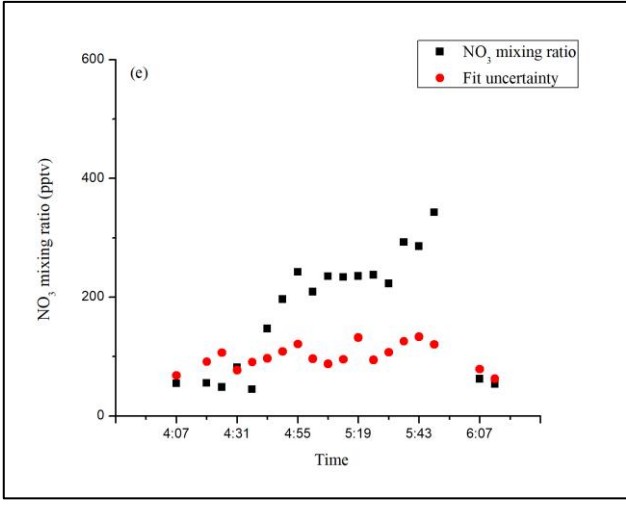





**Figure 5: Time series of NO$_3$ mixing ratios in pptv from 5 nights of measurements (Black squares) with fit uncertainties (Red circles). It may be noticed that the fit uncertainty from analysis was in the range between a low value of ~20 pptv to a high value of ~100 pptv, the variation of which is attributed to variation in aerosol loading**

Time series of the NO$_3$ mixing ratios (black) and uncertainty (red) retrieved from measurements for all five nights are shown in Fig 5. The presence of NO$_3$ is found to diminish immediately on day break every morning. Moderate levels of up to 120 pptv of NO$_3$ were measured on 31$^{st}$ October during the peak activity hours, as shown in fig. 5(a). The highest NO$_3$ concentration of 497 pptv was observed on the night of 5$^{th}$ November (immediately preceding sunrise on 6$^{th}$ morning) as can be seen from fig. 5(b). Figure 5(c) shows NO$_3$ concentration up to 60 pptv on the night of 6$^{th}$ November. On 8$^{th}$ November, the NO$_3$ peak concentration recorded was ~150 pptv and a steady raise after 3:00 A.M. was observed, as shown in fig. 5(d). Figure 5(e) illustrates the NO$_3$ concentration observed during the night of 10$^{th}$ November with a maximum of ~370 pptv.

Several *in-situ* measurements of NO$_3$ in short measurement campaigns have been made previously. We describe below a few of those measurements from different parts of the globe. Brown et al., 2007 made vertical distribution of NO$_3$ measurements in the nocturnal boundary layer in a semi-urban location near Boulder, USA, during 4 – 5 October, 2004. The measurements were made using Cavity Ring Down technique and mixing ratios of above 80 pptv were observed at ~200 m above the ground level.

20 Table 1. Observed NO$_3$ concentration in various locations with corresponding references

| Location | Date | Measured maximum NO$_3$ mixing ratio (pptv) | Method used | Reference |
|---|---|---|---|---|
| Boulder, USA | 4 – 5 October 2004 | < 100 pptv | CRDS | Brown et al. 2007 |
| Beijing, China | February – May 2016 | ~ 50 | IBBCEAS | Wang H et al., 2017 |
| Hebei, China | 28 -30 June 2014 | ~175 | OP-IBBCEAS | Suhail et al., 2018 |
| Thames Estuary, UK | July 2010 | ~200 | IBBCEAS | Kennedy et al., 2011 |
| Brittany, France | September 2006 | < 100 (NO$_3$+N$_2$O$_5$) | IBBCEAS | Langridge et al., 2008 |
| Houston, USA | August – September 2000 | ~ 31 (at sunset) | DOAS | Gayer et al., 2003 |
| Shanghai, China | August – October 2011 | ~95 | DOAS | Wang S et al., 2013 |
| Calicut, India | November 2018 | ~500 | OP-IBBCEAS | This study |

Differential Optical Absorption Spectroscopy (DOAS) was used by Gayer et al., 2003 for measuring daytime NO$_3$ in Houston, USA, as well as by Wang et al., 2013 for nocturnal measurements in Shanghai, China. IBBCEAS technique was used by Wang et al., 2017 in Beijing, China; by Kennedy et al., 2011 in Thames Estuary, UK; Langridge et al., 2008 in Brittany, France, and OP-IBBCEAS by Suhail et al., 2018 in Hebei, China. Many studies have reported a few hundreds of



pptv levels of nocturnal $NO_3$, while our measurements in this study have seen a high of ~0.5 ppbv during one of the nights. In the absence of comprehensive VOC measurements source and sink apportionment is beyond the scope of this study, however, the measurements on $5^{th}$ November showed the highest aerosol loading observed during this campaign coinciding with the highest $NO_3$ mixing ratio observed. Table 1 shows a comparison of previously reported measurements of ambient $NO_3$ in various locations with the present study.

It may be noted that the fit uncertainty levels are different for each night possibly due to the changes in aerosol loading. When the cavity was in open configuration (during measurements after calibration was completed each night) particulate matter (PM) suspended in the atmosphere filled the cavity thereby reducing the effective path lengths [Ruth et al., 2014] to some extent from the anticipated values. While aerosol loading was confirmed from the concurrent measurements of PM mass concentrations with corresponding reduction in our detection sensitivity (increase in uncertainty), we do not expect fluctuations on a time scale faster than our acquisition time as we are not immediately close to any fugitive PM emission sources.

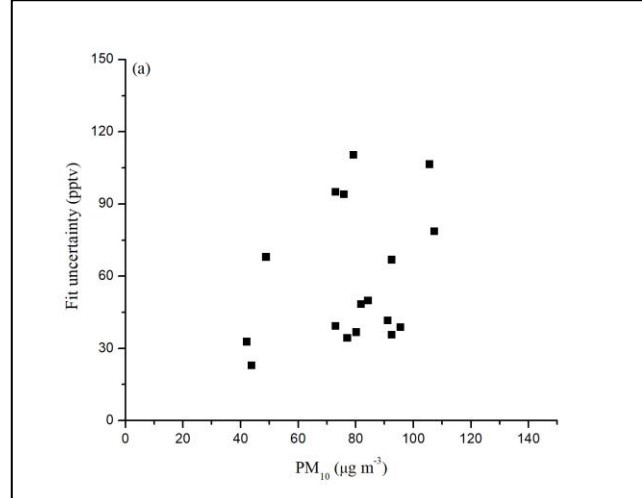
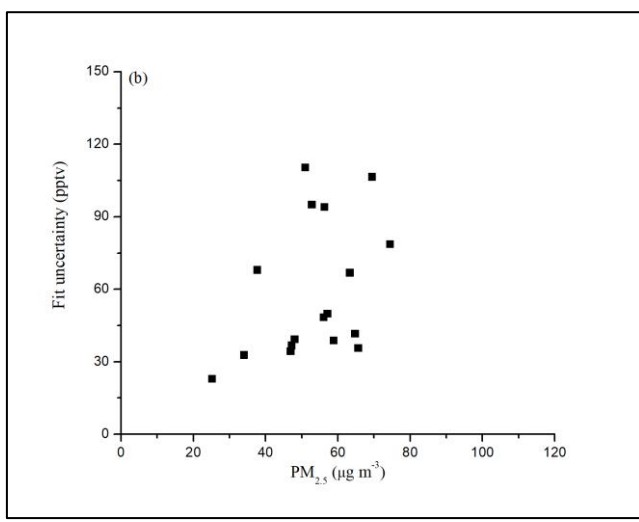

**Figure 6: The scatterplots showing increase in uncertainty with increase in concentration of (a) PM$_{10}$ and (b) PM$_{2.5}$.**

It is to be noted that the measurement of $R$ is made in aerosol free nitrogen atmosphere, giving an effective path length of $d \times (1-R)^{-1}$, where $d$ is the cavity length. The measured reflectivity at 662 nm of ~0.9997 corresponds to an effective path length of ~9.5 km. In a previous study [Suhail et al., 2018], a slightly lower $R$ of ~0.999 provided ~6.5 km effective path length in open-path configuration at a continental semi-urban location in China where moderate aerosol loading were present. The detection limit was ~40 pptv in that study and the variability in aerosol loading was minimal for the duration of measurements when compared to present study. As expected, increased $R$ in the present study has not improved the detection





sensitivity appreciably because the attainable effective path length may have been limited by the atmospheric extinction due to high aerosol loading. In fact, the fit uncertainties were found to be higher during most nights, especially when noticeable increase in PM mass events occurred. The particulate mass concentrations ($PM_{10}$ and $PM_{2.5}$) from concurrent measurements at the location from the nearby observatory (private communication) were used as indicators for aerosol loading and elevated

fit uncertainty events from our instrument were checked against high aerosol loading events. An increase in concentration of either $PM_{10}$ or $PM_{2.5}$ is indicative of an increase in atmospheric aerosol loading at the measurement location. The fit uncertainties were averaged for the same intervals of the hourly reported PM masses and plotted. An increase in the uncertainty with increase in $PM_{10}$ and $PM_{2.5}$ may be inferred from the scatterplots in fig 6(a) and 6(b) respectively. The aerosol present in the atmosphere will be a mixture of species with optical properties ranging from non-absorbing to strongly

absorbing. Therefore, a linear response to aerosol loading of fit uncertainty was not expected. Smaller particles may be dominated by sulphates and nitrates that are non-absorbing in this wavelength range and light extinction due to scattering may be dominant in such a scenario. Studies in similar environments have shown that roughly 40% of PM may be due to sum of elemental and organic carbon [Souza et al., 2014], in which case light extinction due to absorption may have added implication in the fit uncertainty. While a quantitative description of the dependence of the fit uncertainty on aerosol loading

require size measurements, species-wise description of optical properties, and Mie calculations, it is noted that an increase in uncertainty results with general increase in aerosol loading.

**4 Conclusions**

This paper described the use of an OP-IBBCEAS system for *in-situ* detection of the most important nocturnal oxidant $NO_3$ in

a middle-sized urban environment in India due to intense vehicular activities. The nocturnal atmospheric $NO_3$ radical was found to be present in moderate to high mixing ratios in the Indian city of Calicut (Kozhikode) when measured near a vegetable sorting facility and market where diesel vehicles were found idling. An IBBCEAS in open-path configuration (OP-IBBCEAS) was employed for monitoring $NO_3$ radical for a weeklong campaign of which data were available for 5 nights. The instrument was located on roof top of the east wing on Palayam bus terminal in the middle of Calicut city. The activities

in the busy vegetable market started around 3.00 A.M. when truck movements were active. The results obtained from the measurements were consistent to these activities, with high $NO_3$ mixing ratios observed on all nights.

With an open optical cavity of 285 cm the effective mirror reflectivity of ~0.9997 at 662 nm was obtained in aerosol-free nitrogen atmosphere using a low-loss optical window of pre-calibrated loss. This corresponds to an effective pathlength of

~9.5 km. The stability of output intensity of the instrument was found to be excellent throughout the campaign with little need for optical realignments. High to moderate levels of $NO_3$ mixing ratios were found with a maximum of 497 pptv during the busy hours on 5[th] November when aerosol loading was high as well. On an average, the daily NO and $NO_2$ concentrations were found to be minimum at a level of ~20 and ~30 $\mu g\ m^{-3}$ respectively, when $NO_3$ measurements were at the highest levels. The minimum detection limit of the instrument was calculated from the fit uncertainty of the measured



$NO_3$ concentration and was found to vary in the 20 – 50 pptv range. The effect of aerosol loading on the fit uncertainty was studied from the hourly averaged concurrent $PM_{2.5}$ and $PM_{10}$ mass measurements. The fit uncertainty was generally found to be increasing with increased aerosol loading, and has not reduced (to indicate improved sensitivity) with increased $R$ when compared to a previous study [Suhail et al., 2014] in open-path configuration.

Author contribution: K. Suhail participated in planning and experimental activities, data analysis and manuscript preparation. R. Arun carried out the design of the instrument, participated in experiments, data analysis and manuscript preparation. J. Shreya prepared analysis program, carried out the data analysis and participated in manuscript preparation. J. Shebin, S. Aishwarya and P. Anoop participated in the experiment. R. Varma and V. Deepa were involved as investigators as well as participated in experiment planning and general supervision.

Acknowledgements: The authors thankfully acknowledge financial support from the Kerala State Council for Science, Technology and Environment, project code "003/SRSPS/2015/CSTE". We are indebted to Prof. A. A. Ruth for loamimg several components for the experiments. We are also grateful to Mr. Akhil V. M. and Mr. Shafeel K. U. for the assistance received in components fabrication.

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
