# Peer review of "Observation of nocturnal $NO_3$ during vehicular activities in the medium sized city of Calicut in coastal India"

_Atmospheric Measurement Techniques, 2019_

## Referee Comment (RC1) · Anonymous Referee #1 · 11 Feb 2019

The paper describes the application of an IBBCEAS instrument for the detection of the nitrate radical in an urban environment in India. Unfortunately, the paper lacks novelty or substantial progress in instrument technology. The instrument described in this paper has been used in a similar way earlier by the same group of authors and results including the description of the instrument has recently been published by them (Suhail, K., George, M., Chandran S., Varma, R., Venables, D.S., Wang, M., and Chen, J.: Open path incoherent broadband cavity-enhanced measurements of NO3 radical and aerosol extinction in the North China Plain, Spectrochimica Acta Part A, 208, 24-31, doi: 10.1016/j.saa.2018.09.023, 2019.). In addition, NO3 detection by IBBCEAS instruments is an established method that was described by several groups

in the past. No novel approach is shown in this paper. The paper is mixed with the description of results from the application of the instrument in an urban environment close to diesel exhaust emissions. No direct connection between measurements and instrumental questions is shown in the description of the measurements that would be expected for this journal. However, there is also no attempt to explain the plausibility of measurements. It seems rather unlikely that high $NO_3$ concentrations as measured by the instrument here are present close to the emission of diesel exhaust with presumably high NO emissions. Unfortunately, no further details of NOx measurements that were done in the monitoring station close to the site where $NO_3$ was measured are given in order to support results. A discussion about the plausibility of measurements is missing. For these reasons, I recommend to reject the manuscript.

---

## Referee Comment (RC2) · Anonymous Referee #2 · 5 Mar 2019

The manuscript "Observation of NO3 due to vehicular pollution activities in the medium sized city of Calicut in coastal India" by K. Suhail et al, reports on NO3 measurements carried out in the city of Calicut in a polluted environment in autumn 2018 by using OP-IBBCEAS.

The instrumental set up and calibration procedures are well described and referenced and do not significantly differ of those already reported in previous and recent papers of some of the authors, in particular by Suhail et al., Spectrochimica Acta, Part A, 2019.

Apart from the experimental chapter and the verification of the performance of the instrument indicated by instrumental parameters, the analysis of results in the rest of the

manuscript is poor. Provided that the measurement technique and the instrument have already been reported elsewhere, the present work must focus on the interpretation of the NO3 ambient data encountered, and this is definitely insufficiently treated.

Over the text the authors identify themselves most of the deficiencies in the analysis due to measurements and data which are either not available or not shown. The hypotheses are generally supported by qualitative assumptions or partly based on private communications about potential existing information. Furthermore the authors do not attempt to deepen in any explanation of the variability of the NO3 concentrations observed; the relation with vehicular activities although plausible is not really documented.

In summary, I recommend to reject the present manuscript as it remains descriptive and does not provide scientific new findings which deserve publication in AMT. I would encourage the authors to use this work as a preparatory study for a re-designed new campaign with a clear scientific focus and a critical number of simultaneous measurements to enable interpretation.